

# Exceptional sequence of severe thunderstorms and related flash floods in May and June 2016 in Germany. Part I: Meteorological background

D. Piper[1,4], M. Kunz[1,4], F. Ehmele[1,4], S. Mohr[1,4], B. Mühr[1,4], A. Kron[2,4], and J. Daniell[3,4]

[1]Institute of Meteorology and Climate Research (IMK-TRO), Karlsruhe Institute of Technology (KIT), Karlsruhe, Germany
[2]Institute for Water and River Basin Management (IWG), Karlsruhe Institute of Technology (KIT), Karlsruhe, Germany
[3]Geophysical Institute (GPI), Karlsruhe Institute of Technology (KIT), Karlsruhe, Germany
[4]Center for Disaster Management and Risk Reduction Technology (CEDIM), Karlsruhe, Germany

*Correspondence to:* Michael Kunz
(michael.kunz@kit.edu)

**Abstract.** During a 15-day episode from 26 May until 09 June 2016, Germany was affected by an exceptionally large number of severe thunderstorms. Heavy rainfall, related flash floods and creek flooding, hail and tornadoes caused substantial losses running into billions of Euros (EUR). This paper analyzes the key features of the severe thunderstorm episode using extreme value statistics, an aggregated precipitation severity index, and two different objective weather type classification schemes. It is

shown that the thunderstorm episode was caused by the interaction of high moisture content, low thermal stability, weak wind speed, and large-scale lifting by surface lows, persisting over almost 2 weeks due to atmospheric blocking.

For the long-term assessment of the recent thunderstorm episode, we draw comparisons to a 55-years period (1960 – 2014) regarding clusters of convective days with variable length (2 – 15 days) based on precipitation severity, convection-favoring weather patterns and compound events with low stability and weak flow. It is found that clusters with more than eight consecu-

tive convective days are very rare. For example, a 10-day cluster with convective weather patterns prevailing during the recent thunderstorm episode has a probability of only 0.01%.

**Keywords:** flash floods, heavy rainfall, large-scale weather types, severe thunderstorms, convection, persistence

## 1   Introduction

Between end of May and mid of June 2016, Germany as well as large parts of central and southern Europe were affected by

an exceptionally large number of severe convective storms and related extremes such as heavy rainfall, hail, and tornadoes (Fig. 1). Rain totals exceeding 100 mm within few hours at several locations in Germany triggered various flash floods and floods mainly in small catchments. In the town of Braunsbach in the federal state of Baden-Württemberg, for example, a severe flash flood on 29 May with a height of up to 3.5 m caused serious damage to more than 80 buildings, of which five were completely lost (Daniell et al., 2016). Only three days later on 01 June, extreme rain in the district of Rottal-Inn in the south

of Bavaria evoked a sudden and dramatic rise of the levels of several creeks such as the Simbach, where the height increased from 20 cm to more than 5 m within only 12 hours. Subsequently, the village Simbach am Inn experienced the largest flooding



in history. Some of the thunderstorms during the 2 weeks also produced hail with diameters between 0.5 and 5 cm. A total of 12 tornadoes on eight days with intensities between F0 and F1 on the Fujita intensity scale were recorded and confirmed by the European Severe Weather Database (ESWD; Dotzek et al., 2009).

The severe thunderstorms caused substantial damage to buildings, infrastructures, transportation networks and crops. A large number of roads and railroads were blocked or severely damaged, and some villages experienced power outages over a couple of days. Flooded regions such as the district of Rottal-Inn in Bavaria were completely trapped by the water masses and cut off from the outside world. According to MunichRe (2016), the overall losses associated with the severe convective storms in Europe totaled EUR 5.4 bn, of which EUR 2.7 bn were insured. In Germany, economic losses accounted for EUR 2.6 bn with insured losses of EUR 1.2 bn (GDV, 2016).

The large number of severe thunderstorms developed in an environment with moist and unstable air masses that persisted over almost two weeks. A large-scale high pressure system aloft stretching from Great Britain to Iceland and central Scandinavia caused a blocking situation and hampered the exchange of air masses during that episode. The pressure gradient and the resulting wind speed in the lower troposphere were very weak, particularly in the second half of the storm episode. Consequently, thunderstorms were almost stationary, resulting in large precipitation accumulations in local areas.

The objectives of this paper are to highlight the meteorological conditions that were decisive for the thunderstorm episode, to estimate the severity of the recorded rain totals, and to put the event into the historical context. Since information about thunderstorm occurrence is not available over a sufficiently long period, statistical analyses are based on different proxies that estimate the convective potential of the atmosphere: large-scale weather patterns derived from reanalysis data and convective parameters obtained from vertical profiles of radiosoundings. Another purpose of our study is to estimate empirical probability distributions with respect to the variable cluster length of days with convective weather situations using dichotomous parameters such as convection favoring weather types or areal-related precipitation severity index.

While this paper focuses on the meteorological aspects of the severe thunderstorm episode, Part 2 (Daniell et al., 2016) will discuss the impact of selected flash floods in Baden-Württemberg and will present a simplified method to estimate losses from flash floods. All investigation were conducted within the frame of the Forensic Disaster Analysis (FDA) approach in near-real time, which is the main research strategy of the Center of Disaster Management and Risk Reduction Technology (www.cedim.de; Kunz et al., 2013).

The paper is structured as follows: Section 2 presents the different data sets that are used and the methods applied. Section 3 discusses the synoptic background including precipitation observations and the respective probabilities in a long-term perspective. In Sect. 4, we assess the persistence of certain clusters of days with convection-favoring conditions and estimate their occurrence over the last five decades. The last Sect. 5 briefly summarizes the main results and gives some conclusions.

## 2 Data and methods

The episode with an exceptional number of severe thunderstorms extended from 26 May until 09 June 2016 (hereafter referred to as STE16). For the long-term classification of STE16, occurrence probabilities for precipitation, atmospheric stability, and



large-scale weather patterns are assessed with respect to the 55-years period 1960 – 2014 (C20/21) for the summer half year (SHY) from April to September. The study domain for the statistical analyses comprises the whole area of Germany south of 52°N (in case of model data: also adjacent countries), where most of the convective events occurred (cf. Fig. 1).

## 2.1 Observational data

### 2.1.1 Precipitation

Statistical rainfall analyses are based on 24-hour REGNIE totals (*REGionalisierte NIEderschläge*, regionalized precipitation) provided by the German Weather Service (DWD). REGNIE is a gridded data set based on several thousand climate stations (RR collective). Selected station data are interpolated to a regular grid considering elevation, exposition and climatology (Rauthe et al., 2013). The REGNIE area contains 611 grid points in west-east direction with $5.83°E \leq \phi \leq 16°E$ and 971 grid points in north-south direction with $47°N \leq \theta \leq 55.08°N$ ($\phi$: longitude; $\theta$: latitude), but covers only Germany. The spatial resolution is approximately $1\,km^2$. It should be noted that REGNIE data are not homogeneous due to the temporal variability of the number of rain gauges considered.

In addition to REGNIE, we also used data from selected rain gauges of DWD stations with hourly resolution. For each day during STE16, we chose the rain gauge with the highest observed 24-hour total. The observation time period for 24-hour totals (both REGNIE and rain gauges) is from 06 UTC to 06 UTC on the next day, but values are backdated in order to conform with the usual calendar days.

### 2.1.2 Lightning

Lightning data for 2001 – 2014 (SHY) were obtained from the German detection network BLIDS (BLitz-Informations-Dienst Siemens), which is integrated in the European EUCLID (EUropean Cooperation for LIghtning Detection) network (Schulz et al., 2016). We only considered cloud-to-ground flashes (CG) to define convective days without further distinguishing among polarity and peak current. BLIDS data are provided in a spatial resolution of 1 km and a temporal resolution of 1 ms (Drüe et al., 2007).

### 2.1.3 Radiosoundings

Atmospheric conditions prevailing during STE16 and C20/21 were estimated from vertical profiles of temperature, moisture, and wind at four radiosounding stations in western and southern Germany: Essen (51.41°N 6.97°E), Idar-Oberstein (49.70°N 7.33°E), Stuttgart (48.83°N 9.20°E) and Munich (48.24°N 11.55°E). The profiles were provided by the Integrated Global Radiosonde Archive (IGRA) from the National Climatic Data Center (Durre et al., 2006). For the assessment of thermal stability, we used the surface-based lifted index (SLI) and the convective available potential energy (CAPE). Both quantities have been identified in various studies to well represent atmospheric stability (Davis et al., 1997; Haklander and van Delden, 2003; Manzato, 2003; Kunz, 2007; Mohr and Kunz, 2013).



Since the movement of thunderstorms to a large degree is controlled by the wind vector at mid-tropospheric levels (depending on the vertical extent of the cell), we also considered wind speed and direction at 500 hPa, respectively.

## 2.2 Model data

The study domain for the model data is slightly extended to the border regions and covers the area of $5.5°E \leq \phi \leq 15.0°E$, $47.5°N \leq \theta \leq 52.0°N$. Only data sets at 12 UTC ($\sim$ 12:38 true local time) were considered, since they best mirror the prevailing convective conditions. Model data were used to estimate the large-scale weather situation during both STE16 and C20/21.

### 2.2.1 CoastDat2

The CoastDat2 reanalysis was employed here for investigating long-term convection-favoring conditions. The reanalysis has been carried out by the Helmholtz-Zentrum Geesthacht – Centre for Materials and Coastal Research in Germany (Geyer and Rockel, 2013; Geyer, 2014). CoastDat2 is based on the COSMO (Consortium for Small-Scale Modelling) model in climate mode, COSMO-CLM Version 4.8 (Rockel et al., 2008), and uses the National Centers for Environmental Prediction/National Center for Atmospheric Research reanalysis (NCEP/NCAR1; Kalnay et al., 1996) as forcing, which in consequence of the almost constant data assimilation only exhibits a small trend. The model output is available for the entire European domain in a resolution of 0.22° with 40 vertical model layers from 1948 (including 3 years spin-up time) until today.

### 2.2.2 CFSv2 Operational Analysis

Since CoastDat2 data are not available for STE16 yet, we estimated prevailing weather patterns from Climate Forecast System (CFSv2) Operational Analysis data, which has been in operation since April 2011 as the successor of the Climate Forecast System Reanalysis (CFSR, Saha et al., 2010, 2014). The CFSv2 data are produced under guidance of the National Centers for Environmental Prediction (NCEP) and offer hourly data with a global horizontal resolution of 0.5°.

## 2.3 Objective weather types of DWD

The objective weather type classification (OWLK) designed by DWD (Dittmann, 1995; Bissolli and Dittmann, 2001) differentiates between 40 weather patterns by quantifying three model parameters in a dichotomous scheme: (i) mean flow direction $AA$ at 700 hPa with the possibilities of SW, NW, NE, SE, and an indefinite type XX, (ii) cyclonality $CY$ as the product of geostrophic vorticity and Coriolis parameter at low (950 hPa) and mid-tropospheric levels (500 hPa), yielding either cyclonic (C) or anticyclonic (A) flow, and (iii) humidity index $HI$ represented by the precipitable water with the climatological daily average removed; positive/negative anomalies are denoted by M (moist) and D (dry).

The continuous grid point values contained in the reference domain are converted into one scalar number for each day and each parameter, which is mapped on a categorical variable in each case using previously defined thresholds. The four variables are concatenated forming a character code

$$AA \, CY_{950\,hPa} \, CY_{1000\,hPa} \, HI \tag{1}$$



For example, the pattern SWCAM refers to a mainly southwesterly flow, cyclonic/anticyclonic at 950 hPa / 500 hPa, respectively, and a higher moisture content compared to climatology.

Grid points near the center of the domain are weighted by a factor of three, those located near the margins by a factor of one and all points in an interjacent zone by a factor of two, so as to restrict the influence of the outer areas. In this paper, the classification results obtained by DWD are used, which rely on the reference domain defined by Dittmann (1995) comprising Germany and parts of the neighboring countries.

Several studies have established a relationship beetween specific OWLK types and convective activity in terms of severe hailstorms (Kapsch et al., 2012; Mohr et al., 2015) or tornadoes (Bissolli et al., 2007). The advantages of OWLK compared to subjective methods are the non-ambiguous assignment criteria and the automated categorization procedure (Philipp et al., 2010).

## 2.4 Convective weather types

It can be shown that OWLK only has a limited skill regarding the identification of ambient conditions favorable for convective activity, since it mainly aims at classifying the synoptic situation in general. Based on the methodical approach of OWLK, we therefore developed a new objective weather type classification (conOWLK) with a special focus on convection. This scheme consists of four parameters: equivalent potential temperature at 850 hPa, precipitable water, surface-based lifted index (SLI) and vertical velocity $w$ at 500 hPa. In the former two cases, we removed the average annual cycle by subtracting the 10-day running mean over the average daily values. The parameter values for a respective day are obtained analogously to OWLK by calculating the weighted areal mean over a rectangle now enclosing the study domain defined in Sect. 2.2.

The continuous variables are transformed into discrete ones using trichotomous parameters instead of dichotomous ones as for OWKL, allowing for those values that can not be allocated clearly to one of the two original classes being comprised by a third, neutral class (abbreviated as X). The thresholds are determined so as to distinguish best between conditions favoring and inhibiting convection, which is assessed by categorical verification with respect to lightning data. For this purpose, we calculated the distribution of the Heidke Skill Score (HSS) on a large range of possible threshold values for each parameter separately and chose those two values as thresholds, where HSS equals its 90%-quantile.

Convection-favoring weather types among the total of 81 classes were verified and identified against convective days according to BLIDS data using categorical verification. These thunderstorm-related types are characterized by relatively warm (W) and moist (M) conditions with instability (I) and either lifting (L) or no vertical motion (X) present, yielding the two codes WMIL and WMIX. Conversely, the two weather types inhibiting convection are given by the codes CDSS and CDSX (cold, dry, stable and subsidence / no vertical motion). All other types are referred to as neutral.

Since the objective of conOWLK is to identify ambient conditions that imply a very high chance for the development of convection, a significant amount of events are missed. Therefore, it is reasonable to additionally implement a less strict classification. This is done by developing a multivariate statistical model based on quadratic discriminant analysis (qdaOWLK) assigning a particular day to one of the groups *convective day (Yes/No)* depending on the values of several continuous input variables. These variables are also represented by the four parameters used in conOWLK.





First, qdaOWLK is calibrated using CoastDat2 reanalysis and lightning data (2001 – 2014). These training data can be divided into two subsets corresponding to the groups of *convective* and *non-convective* days, which are characterized by two different multivariate probability density distributions (Marinell, 1998). Based on this partitioning, an assignment rule in terms of a discriminant function is developed, which will be used to classify those days, when lightning data are not available.

Since covariance matrices differ significantly between both subsets (heteroscedasticity), quadratic instead of linear discriminant analysis has to be performed (Sánchez et al., 1998). Applying a Kolmogorov-Smirnov test to the time series of the four parameters yields significant deviations from the normal distribution. Therefore, data are normalized using a Yeo-Johnson power transformation (Yeo and Johnson, 2000). In this study, the quadratic discriminant function $\delta$ is derived using a maximum likelihood criterion (Sánchez et al., 1998). Hence, an arbitrary entity is assigned to the group exhibiting the higher value of the

likelihood function $L_m$ with $m \in \{0,1\}$ corresponding to the populations of non-convective and convective days, respectively.

It can be shown that $\delta$ is computed from the input data vector $\boldsymbol{x}$ for a particular day by:

$$\delta = \frac{1}{2}\boldsymbol{x}^T(\Sigma_0^{-1} - \Sigma_1^{-1})\boldsymbol{x} + \boldsymbol{x}^T(\Sigma_1^{-1}\boldsymbol{\mu_1} - \Sigma_0^{-1}\boldsymbol{\mu_0})$$
$$+ \frac{1}{2}\boldsymbol{\mu_0}^T\Sigma_0^{-1}\boldsymbol{\mu_0} - \frac{1}{2}\boldsymbol{\mu_1}^T\Sigma_1^{-1}\boldsymbol{\mu_1} + \frac{1}{2}\ln\left(\frac{|\Sigma_0|}{|\Sigma_1|}\right), \tag{2}$$

where $\Sigma_m$ represents the covariance matrix of population $m$, the superscript $-1$ denotes the inverse matrix and $\mu_m$ is the

respective sample mean vector. Due to

$$\delta = \ln[L_1(\boldsymbol{x})] \ - \ \ln[L_0(\boldsymbol{x})], \tag{3}$$

an arbitrary day is classified as convective if $\delta > 0$.

Model performance is assessed by means of leave-one-out cross validation (Wilks, 1995). Here, the discriminant function is computed from the sample of training data excluding the first day, which then is classified by the model. For a sample of

size $n$ this procedure is conducted $n$ times shifting the day excluded by one step each time. As a result, the $n$ predictions for *convective day (Yes/No)* are compared with the actual incidences using categorical verification.

## 2.5   Return periods

To estimate statistical return periods of precipitation totals $R$, we applied the classical generalized extreme value (GEV) distribution. Most appropriate for precipitation statistics is the Fisher-Tippett type I (also known as Gumbel) distribution (Wilks,

1995), with a cumulative distribution function (CDF) of:

$$F(R) = \exp\left[-\exp\left(\frac{\zeta - R}{\beta}\right)\right], \tag{4}$$

where $\beta$ and $\zeta$ are scale and location parameter, respectively. The two free parameters of the CDF are estimated by the Method of Moments (Fisher and Tippett, 1928; Gumbel, 1958) using annual rainfall maxima during C20/21:

$$\beta = \frac{\sigma\sqrt{6}}{\pi}; \qquad \zeta = \bar{R} - \gamma \cdot \beta, \tag{5}$$



where $\bar{R}$ is the sample mean, $\sigma$ the sample standard derivation and $\gamma$ the Euler-Mascheroni constant ($\approx 0.5772$).

The CDF describes the probability of occurrence $P$ of a value $R$ beneath a threshold $R_{\text{trs}}$: $F(R) = P(R < R_{\text{trs}})$. On the other hand, the return period $t_{\text{RP}}$ is related to the probability of threshold exceedance $P(R \geq R_{\text{trs}}) = t_{\text{RP}}^{-1}$. Therefore, the CDF can be written as $F(R) = 1 - t_{\text{RP}}^{-1}$. The resulting equation for the return period $t_{\text{RP}}$ is:

$$t_{\text{RP}}(R) = \left[ 1 - \exp\left( -\exp\left( \frac{\zeta - R}{\beta} \right) \right) \right]^{-1}. \tag{6}$$

## 2.6 Heavy rainfall and Precipitation Severity Index

Heavy rainfall usually is defined either by the exceedance of appropriate percentiles (e.g., 99 or 99.9%; see below) or using a fixed threshold as a function of duration. In the latter case, we considered the criterion according to Wussow (1922):

$$N_{\text{cr}} = \sqrt{5 \cdot D} \tag{7}$$

with the critical rain rate $N_{\text{cr}}$ representing a threshold for heavy convective rainfall with duration $D$ (in minutes) between 30 minutes and 24 hours.

The annual variability and the persistence of extreme rain events over past decades are examined by using the precipitation severity index $PS$, which is an aggregate measure of both the intensity and the spatial extent (see also Schröter et al., 2015):

$$PS_\xi^k = \frac{1}{\Gamma} \sum_{i,j} \left( R_{i,j}^k \right) \mid R_{i,j}^k \geq R_{i,j}^\xi, \tag{8}$$

where $R_{i,j}$ represent the rain totals at REGNIE grid points $(i,j)$, $k$ denotes a certain day, $\Gamma$ is the size of the investigation area, and $\xi$ are the 99 or 99.9% percentiles of the distribution function quantified independently at all grid points during C20/21. In this formulation, all totals are accumulated that are equal to or exceed the value of the 99 or 99.9% percentile, respectively, at the respective points. Normalization of the accumulated totals by the total area in Equation 8 gives the dimensionless precipitation severity index $PS$.

Whereas the 99% percentile (1.83th largest total in a year on average) may contain advective precipitation as well, the 99.9% (0.18th largest total) mainly considers heavy convective rainfall. Within the investigation area, the 99% (99.9%) percentiles vary between 15 mm (30 mm) over the northern lowlands and almost 60 mm (100 mm) over the peaks of the German Alps and the Black Forest Mountains (not shown).

## 2.7 Persistence analysis

Days with widespread thunderstorms tend to form temporal clusters of variable length, which can be described statistically by the concept of persistence. In this study, a persistent cluster is defined as a sequence of days (between 1 and 15 days) with the binary parameter taking the value of 1 (event day) or 0 (non event day). Clusters with a length of up to seven (15) days may contain at most one day (two days), on which the event does not occur (skip day). Each time, the algorithm identifies the end of a cluster according to these rules, the actual length exclusive of skip days is stored. The described approach is a top-down one: It considers only the maximum cluster length and prevents that longer clusters are split into two or more sub-clusters. Dividing





the absolute number of clusters with length $n$ by the number of all clusters yields the relative frequency of a cluster with length $n$. Due to the large data volume, this can be perceived as an approximate measure for probability.

Persistence analysis as described above is applied to precipitation severity index $PS_\xi$ (Eq. 8), large-scale weather types, and compound events with low stability and weak flow.

## 3   Weather situation

Warm and moist air masses in combination with large-scale lifting by shallow surface lows persisted during STE16 over wide parts of Germany. Steep environmental lapse rates due to surface heating by solar radiation, cooling at mid and upper troposphere levels by cold air advection, and upper level troughs created an environment favoring the development of various thunderstorms. Due to the very weak horizontal flow at mid-tropospheric levels – particularly in the second half of the period – the various thunderstorms were almost stationary, resulting in large precipitation accumulations over limited areas.

### 3.1   Synoptic overview and atmospheric characteristics

At the end of May and beginning of June, large parts of Europe were influenced by atmospheric blocking. Such a blocking event is characterized by large amplitude, long-lasting, and negative potential vorticity anomalies located beneath the dynamical tropopause (Croci-Maspoli et al., 2007). They establish most frequently over the North Atlantic and the eastern North Pacific during all four seasons, but most frequently in autumn and winter. The usual westerly flow over Europe is blocked due to the presence of a high pressure system over the North Atlantic or northern Europe. Such a block, termed to as *Omega*-block in Europe, may persist over several days up to several weeks with enormous consequences for the regional weather and climate (Masato et al., 2013).

During STE16, the structure of the 500 hPa mean geopotential height over Europe and North Atlantic was characterized by a massive high pressure system, which stretched from Great Britain to Iceland and central Scandinavia. The high pressure block was flanked by two upper level troughs (Fig. 2a). To the west a trough made its way southwards towards the Azores, whereas over eastern Europe another trough extended southwards to the Black Sea and Turkey.

The 500 hPa geopotential anomalies with respect to the long term mean (1979 – 2005) show some positive anomalies of around 20 hPa occurring near Iceland, whereas negative anomalies of the same magnitude were present in the Mid-Atlantic Ocean around the Azores (Fig. 2b). Unlike to usual blocking situations, an area of low pressure aloft was cut off beneath the northward bulging high pressure ridge. Low 500 hPa geopotential values can be identified over Poland, Germany and France, accompanied by weak pressure gradients resulting in low wind speeds at mid-tropospheric levels. The weak upper-level trough over Central Europe corresponded with shallow surface lows, one of which extended between Poland and France and persisted until 09 June. Consequently, western and southern Germany remained under the influence of low pressure with moist and warm air, while drier air gradually prevailed in the northeast. At the beginning of the blocking event, moist and warm air was advected ahead of a deep trough northeastwards towards Central Europe. Later on in June with the quasi-stationary low pressure being



present across Central Europe, moisture was maintained mainly by evapotranspiration from local sources and advection from nearby countries.

The most intense and fatal rain events occurred on 29 May over southwestern Germany. In the town of Braunsbach, for example, flash floods and landslides had devastating consequences (see Fig. 4a). During that day, the upper level trough approached southern Germany from France and Switzerland. It was associated with massive positive vorticity advection and the advection of warm and moist air at mid and lower tropospheric levels, which both reached their maximum values during the evening hours. Together, they provided a large uplift across western Bavaria and Baden-Württemberg. The preexisting airmass was unstably stratified leading to negative values of SLI of $-5$ and $-2$ K in Munich and Stuttgart, respectively (Fig. 3).

First thunderstorms on that day occurred over Bavaria already before noon. While the rain area extended towards western Baden-Württemberg, new and intense thunderstorms formed north of the Alps, followed by isolated heavy thunderstorms aligned from Munich to Salzburg and Nuremberg to Stuttgart. In the evening, a large Mesoscale Convective System (MCS) Type II, which usually develops from preexisting single cells, covered all of Baden-Württemberg, western Bavaria, eastern Rhineland-Palatinate and southern Hesse with a size of roughly $60.000\,\mathrm{km^2}$. At 19 UTC, a line of violent thunderstorms stretched over several hundred kilometers from Passau in eastern Bavaria to Mannheim in the north western tip of Baden-Württemberg. Various convective cells embedded in the eastern and northern edge of the MCS affected mainly the same region in the north of Baden-Württemberg leading locally to rainfall totals in excess of 100 mm within few hours (see Fig. 4a).

During the entire STE16 period, atmospheric stability across Germany (and Central Europe) was low. The SLI computed at four sounding stations shows values below zero on almost every day (Fig. 3a; recall that negative SLI values express instability). SLI was particularly low at the beginning and in the second half of STE16. The values for $\mathrm{CAPE_{100hPa}}$, which is calculated based on start values of the lifted curve being mixed over the lowest 100 hPa, were above $400\,\mathrm{J\,kg^{-1}}$ on most days with maximum values between 800 and $1100\,\mathrm{J\,kg^{-1}}$.

Apart from the first two days, wind speed at 500 hPa was exceptionally low with values between only 2 and $15\,\mathrm{m\,s^{-1}}$ (Fig. 3b). Especially at the two stations located in southern Germany, Stuttgart and Munich, where also most of the flash floods occurred, the values dropped below $10\,\mathrm{m\,s^{-1}}$ on 31 May and remained on a very low level until the end of STE16. Only the two stations situated in the center of Germany showed wind speeds of around $10\,\mathrm{m\,s^{-1}}$ or slightly above. The general wind direction in 500 hPa until 28 May was predominantly from the west at all four stations considered. Afterwards, the flow turned to southerly (29/30 May) and mainly easterly directions that prevailed until 06 June. The last three days were again dominated by westerly flow. Note, however, that wind directions during calm winds have a very limited applicability.

According to DWD analysis (DWD, 2016), the OWLK pattern that prevailed on the first three days (26 to 28 May) was SWCCM, indicating moist southwesterly flow with cyclonic rotation at both levels (Table 1). Several studies identified this pattern to be most related to severe thunderstorm occurrence in Germany (Kapsch et al., 2012; Mohr et al., 2015). After the first three days, flow direction became mainly indefinite (XX) due to the very weak winds connected to the low pressure gradients. On all days, atmospheric moisture was increased, yielding the weather types XXAAM and XXCCM. These two types have been found to promote thunderstorms as well, for example with a probability of 10% for damaging hail (Kapsch et al., 2012).



As discussed in Sect. 2.4, the original OWLK has not been designed especially for the analysis of convective conditions. Therefore, we developed a new classification (conOWLK) optimized in this regard. According to conOWLK and using CFSv2 reanalysis data (Sect. 2.2.2), 6 days can be classified as convection-favoring (WMIL/WMIX). These two weather types are characterized by near-zero false alarm rates, but a considerable number of missed events with respect to the occurrence of

convective days (not shown). This feature is equivalent to the statement that a categorization as WMIL or WMIX is by approximation sufficient, but not necessary for the actual incidence of lightning. Due to this strict design of conOWLK, WMIL and WMIX coincide with a very high probability for the development of severe thunderstorms. Thus, the number of 6 out of 15 days exhibiting one of these weather types has to be considered as fairly high.

## 3.2 Precipitation

The majority of thunderstorms that developed during STE16 showed a typical diurnal cycle peaking in the afternoon and early evening. On most of the days, intense rainfall affected only small areas with extensions of only few square kilometers. According to radar data, the diameters of the cloud bursts were in the order of several hundreds of meters to 1 or 2 kilometers.

### 3.2.1 Rainfall totals

Thunderstorms with at least 50 mm rainfall totals within 24-hours occurred on 11 days during STE16 (Table 2). Most of the

rain fell within only few hours. As shown in Table 2 for selected rain gauges, the Wussov criterion (Eq. 7) is met on seven of the eight chosen stations, for which hourly data are available. The largest rain amount during this period was observed at Gundelsheim ($\sim$ 50 km west of Braunsbach) with 122.1 mm on 29 May 2016. On this day, heavy rainfall associated with the large MCS caused widespread totals in excess of 50 mm (see also Fig. 4a). Around Simbach in Bavaria, a maximum of 74.6 mm was recorded on 31 May, triggering another devastating flood. Unfortunately, the rain gauge had a malfunction on

01 June during four hours. Filling up the gap with DWD radar data gives a daily total of about 120 mm on this day, which may be even higher than the given value in Table 2. Hence, Simbach would be the only station with a Germany-wide maximum on two days during STE16.

Accumulated 15-day rain totals in Germany widely exceeded a height of 100 mm (Fig. 4b). In several parts of Baden-Württemberg and Bavaria even more than 250 mm were observed, which is by far more than the climatological mean for

the entire month of June. Dry conditions prevailed only towards the North Sea and Baltic region as well as in most parts of Brandenburg, where values rarely exceeded 40 mm.

### 3.2.2 Return periods

Return periods estimated with respect to C20/21 allow for assessing the observed totals in the historic context. Based on Eq. 6, we determined return periods for each day and each grid point during STE16 based on REGNIE 24-hour totals. Additionally,

we calculated 7-day accumulated totals and corresponding return periods for each of the 7-day intervals contained in STE16 and shifted against each other by one day, respectively. Afterwards, we identified the highest return period at each grid point,





both for 24-hour and 7-day totals. These results together with the days / time intervals of maximum return periods are shown in Fig. 5.

Daily totals reveal widespread return periods larger than 40 years, but with several hot spots of more than 200 years, especially around Braunsbach and the far west of Germany (Fig. 5a). The temporal distribution (Fig. 5b) shows that 29 May, and to
a less degree 01 June represent the dominant days for these totals. Minor return periods between 40 and 60 years occurred in the Simbach area, which can be explained by the malfunction of the most important rain gauge in that area on 01 June.

As shown in Table 2, most of the observed daily maxima recorded at selected rain gauges had return periods of at least 5 years with the two outstanding events of 29 May (Gundelsheim) and 01 June (Hamminkeln-Mühlenrott) exhibiting values of more than 200 years. The reconstructed data at Simbach on 01 June would yield high return periods of about 140 years as well.
Considering the 7-day totals, a westward shift of the affected area is obvious. Emphasis now is Rhineland-Palatinate west of Frankfurt (Fig. 5c), again with enclosed areas of very high return periods including the upper catchment of river Ahr (along the border between Rhineland-Palatinate and North Rhine-Westphalia). The majority of the maximum return periods occurred during the first 7-day period from 26 May to 01 June (Fig. 5d, pink), leading to the most serious flood along river Ahr ever reported. Note that 24-hour totals in some regions were such exceptionally high that they also massively affected the 7-day
totals and corresponding return periods.

Comparing the spatial distributions of 24-hour and 7-day return periods (and additionally of 3-day and 14-day periods, which are not shown), it can be concluded that in most regions heavy precipitation occurred just on one or two days during STE16 (e. g. Braunsbach or Simbach). Only few regions such as Rhineland-Palatinate, especially the upper Ahr catchment, experienced rainfall during more than two days. This fact is a typical feature of heavy convective rainfall, which was more or
less randomly spread over large parts of Germany during STE16.

## 4   Persistence analysis

Besides exceptional rainfall totals with return periods in excess of 200 years, another peculiarity of STE16 was the almost daily occurrence of severe thunderstorms somewhere in the investigation area. For this reason, we investigate how often persistent clusters of days with convective weather conditions occur in the long-term mean (C20/21) in terms of heavy rain events based
on the precipitation severity index $PS_\xi$ (Sect. 4.1), large-scale weather patterns (Sect. 4.2), and compound events with low stability and weak flow (Sect. 4.3).

### 4.1   Heavy rainfall

The time series of the annually-accumulated precipitation severity index $PS_\xi^k$ (Eq. 8) for the two percentiles 99 and 99.9% based on REGNIE totals are qualitatively similar (Fig. 6). In both cases, the temporal variability is high with values in a
range between 14,865 and 116,983 for $PS_{99}$ and between 432 and 36,125 for $PS_{99.9}$. An interesting feature is the long-term oscillation inherent in both time series, but more pronounced in the case of the convection-dominated index $PS_{99.9}$. The FFT power spectrum (removal of linear trend) reveals the highest peaks for a periodicity between 2 and 3 years as well as a large



peak for 13 years (not shown). Possible reasons for these oscillations remain unclear, since a direct link to weather patterns or stability cannot be established. The two time series in Fig. 6 also show positive (linear) trends, which, however, are statistically not significant ($\alpha > 5\%$) due to the large volatility.

During STE16, $PS_{99.9} > 0$ at any location (REGNIE grid point) in the investigation area occurred on 10 consecutive days, whereas $PS_{99} > 0$ was reached even on 14 days. To assess the exceptional nature of this persistence, we estimated the occurrence probability of clusters with a length between 2 and 15 extreme rainfall days during C20/21 (cf. Sect. 2.7). Extreme rainfall days are defined when $PS_\xi > 0$ at any location across the investigation area, which best represents the spatial characteristics of STE16 with spatially varying hotspots of heavy rainfall.

According to Fig. 7, the occurrence probability of a 14-day cluster with $PS_{99} > 0$ is 0.56%. Aside from the year of 2016, this cluster occurred only 4 times during C20/21 (1970, 1997, 2000, and 2006). For $PS_{99.9} > 0$, the maximum cluster had a length of 10 days, and occurred 3 times in C20/21 (1963, 1972, and 2002; see Fig. 8). Thus, the probability of such a cluster is even lower with a value of 0.30%.

Figure 7 also shows that on most of the days, $PS_\xi$ is not exceeded in the entire investigation area (60.0% of all days for the 99, and 83.9% for the 99.9% percentile). As expected, the relative frequency of occurrence substantially decreases with increasing cluster length, but with some exceptions. For example for $PS_{99.9}$, a cluster length with 10 days has a higher probability than a cluster of only 9 days (0.3% vs. 0.18%, or 2 vs. 3 events, respectively). This apparently counterintuitive behavior can also be observed in the probabilities of the clusters for the weather types and the compound low stability and weak flow events (cf. Figs. 9 and 12). In both cases, however, the changes affect different cluster lengths. The reason for this behavior remains unclear, but might be the consequence of the large natural variability of convective weather as already indicated by the large volatility of $PS_\xi$.

The time series of the different clusters do not reveal any systematics; rather their occurrence has a large stochastic component (Fig. 8). However, whereas the number of shorter clusters ($2 - 5$ days) increased during C20/21, the number of larger clusters ($> 5$ days) slightly decreased. The lower number of larger clusters cannot compensate the increase of shorter clusters, leading to an overall positive trend. However, due to the large annual variability and the small number of clusters, a robust statement about an increase or decrease cannot be derived from this analysis.

## 4.2 Large-scale weather types

Considering the original OWLK provided by DWD, a cluster length of 8 days (1 skip day) regarding the weather types XX..M (where . denotes either cyclonal (C) or anticyclonal (A) vorticity in each case) was observed during STE16 (cf. Table 1). Including the first three days exhibiting the weather type SWCCM, which has been shown to favor thunderstorm formation as well (Sect. 3.1), even yields an 11-days cluster (2 skip days). Both lengths have never occurred before since the beginning of the OWLK record in 1979.

The total of 6 convective days according to conOWLK in the investigation area subdivide into three clusters of lengths three, two and one on 28/29 May, $03 - 05$ June and 08 June, respectively (Table 1). It is interesting to note that these clusters are separated by several days assigned to the group of neutral weather types. This finding can be attributed to the concept of





combining single trichotomous parameters. Recall that conOWLK classifies a day as neutral or convection-inhibiting if just one of the parameters is slightly below the considered threshold, whereas the other three parameters may be well above.

As already discussed in Sect. 3.1, only days with a very high probability for strong convective development will be classified as convection-favoring concerning all four parameters. Except for the first and last day, when the convective situation was not fully developed, all days were categorized as warm, and all but 2 days were assigned to the moist and unstable class. In contrast, 6 days with large-scale subsidence ($w < 0$) were detected. Consequently, neglecting the lifting parameter $w$ would yield two clusters of 4 days and one cluster of only 1 day. Since STE16 was characterized by spatially varying hotspots of severe convective activity, it can be assumed that convection was mainly triggered by meso-scale flow convergence in the boundary layer instead of large-scale lifting – except for the large MCS on 29 May. Thus, strong local vertical velocity maxima may be overcompensated by subsidence in other regions.

These findings suggest that conOWLK is too strict when persistence analysis with respect to the presence of a largely high convective predisposition is attempted instead of focusing on days characterized by very high probabilities of strong thunderstorm events only. Therefore, another type of classification abbreviated as qdaOWLK is used, which does not rely on a simple combination of parameters exceeding the respective thresholds. Quadratic discriminant analysis as introduced in Sect. 2.4, provides a suitable statistical model that is based on the parameter values calculated by the conOWLK algorithm as well, but leaves out mapping these parameters on categorical variables. Cross validation with respect to the occurrence of convective days yields a HSS of 0.53. According to this model and using CFSv2 data, a cluster length of 11 convective days was detected during STE16 (2 skip days).

As for the precipitation severity index $PS_\xi$, we also estimate the empirical relative frequency distribution of different cluster lengths with respect to qdaOWLK. For this purpose, a time series of the binary variable *convective day* was computed for C20/21 using the discriminant model. As shown in Fig. 9, two thirds of all clusters exhibit a length of one or two. Cluster durations of more than 15 days have not occurred yet since the beginning of the reference period. Less than one percent of all cluster lengths exceed ten days. Consequently, the eleven day cluster of convective days has to be considered as highly exceptional. Due to the sample size of 811 clusters, the relative frequencies obtained can be interpreted by approximation as probabilities.

All cluster lengths exhibit a strong interannual variability (Fig. 10), which is characterized by a large stochastic component, as was the case regarding days with heavy rainfall (Fig. 8). In some years, we observe remarkable extrema, for example in 1989, when 14 clusters of length $2 - 3$ and only 1 of length $4 - 5$ occurred. Conversely, 1991 generally was characterized by weak convective activity (2 clusters with $2 - 3$ days and 1 cluster with $4 - 5$ days). A 10-11-days cluster occurred only 5 times (1964, 1972, 1977, 1994, 1997). In contrast to the time series of heavy rainfall clusters, no long-term increases or decreases are visible.





### 4.3 Compound events with low stability and weak flow

In the last step, we investigate compound events with low stability and weak flow prevailing during STE16 with the same methods as applied above. The following two threshold combinations based on Surface lifted index (SLI) and horizontal wind speed in $500\,\mathrm{hPa}$ ($v_{\mathrm{H,500hPa}}$) are considered in the analysis:

1. Basic criterion (BC):

$$\mathrm{SLI} < 0\,\mathrm{K} \quad \text{and} \quad v_{\mathrm{H,500hPa}} < 10\,\mathrm{m\,s^{-1}},$$

   2. Strict criterion (SC):

$$\mathrm{SLI} < -1.3\,\mathrm{K} \quad \text{and} \quad v_{\mathrm{H,500hPa}} < 8\,\mathrm{m\,s^{-1}}.$$

The thresholds representing the BC are defined by the maximum of the daily minima at all four sounding stations during STE16
(see Fig. 3), representing the prevailing atmospheric conditions during STE16 in the investigation area. In that period, the BC criterion was fulfilled on 13 days (sounding stations Stuttgart and/or Munich; see Fig. 11a). The SC is defined in the same way as BC, but neglecting the day with the highest values; it was fulfilled at the station of Munich with a cluster length of 10 days (two skip days; see Fig. 11b)

Based on sounding measurements and reanalysis data (CoastDat2), we investigated the frequency of varying cluster lengths
for both criteria (BC/SC) during C20/21. According to Fig. 12, the relative frequency of cluster lengths greater or equal than 13 days (BC) or 10 days (SC), respectively, is very low ($< 1\%$). This applies to both the analysis based on the Stuttgart sounding as well as to that considering ($3 \times 3$) grid point averages from CoastDat2 near Stuttgart and Munich. Regarding wind speed $v_{\mathrm{H,500hPa}}$ solely, an almost identical behavior is found. It has to be noted that longer cluster lengths for BC/SC combinations based on sounding data occur less frequently compared to those based on model data, which means that those events are
probably overestimated in CoastDat2.

The climatological distributions of the BC/SC combinations show a distinct north-to-south gradient (not shown). Therefore, regions in the south of Germany show a higher frequency of both the mean distribution of days with compound low stability and weak flow and events with longer cluster lengths (cf. Stuttgart vs. Munich in Fig. 12). This fact can be explained by the lower stability in southern Germany in the mean (Mohr and Kunz, 2013).

Finally, we examined the spatial extent of the SLI/wind combinations in the investigation area. In the first step, we identified the area that fulfilled the BC/SC combinations on each day during STE16 based on CFSv2 analysis. Accordingly, both criteria were reached in an area of at least $A_c = 14 \cdot 10^4 \,\mathrm{km}^2$. This value is considered as a lower threshold for the convective area. In the second step, now based on CoastDat2, we checked for each day during C20/21 whether the BC/SC combination was fulfilled over an area of at least $A_c$ or not. The persistence analysis of that time series is also shown in Fig. 12. For both BC/SC
combinations, the relative frequency is higher compared to the persistence without considering the spatial extent (Stuttgart and Munich). For example, the probability of a cluster length greater or equal than 13/10 days is between 3.0 and 3.5% for BC/SC. Finally, we identified the total area with $2.7 \cdot 10^6 \,\mathrm{km}^2$, where these compound events (according to the BC criterion) prevailed





during the 13 days in STE16. This affected area has been unique until now and was never reached in C20/21, where the area normally was between $0.8$ and $1.9 \cdot 10^6$ km$^2$ for the same (or even higher) cluster length.

## 5   Conclusions

The severe thunderstorm episode in May/June 2016 in Germany (STE16) has been investigated with respect to rain intensity
and the presence of convection-favoring conditions in comparison to a 55-year control period C20/21 ($1960 - 2014$). For the latter, we considered different proxies such as convective parameters obtained from soundings and large-scale weather patterns computed from reanalysis data. We estimated empirical probability distributions with respect to variable cluster lengths of consecutive days with convective weather situations based on the different proxies.

The results illustrate that the interaction of convection-favoring weather patterns, low thermal stability, and weak wind speed
provided important boundary conditions for the extraordinary thunderstorm anomaly observed. Due to atmospheric blocking, these conditions persisted over almost two weeks. The low wind speed at mid-tropospheric levels ensured that convective cells were almost stationary, leading to locally extreme rain accumulations of more than 100 mm, yielding return periods in excess of 200 years for both 24-hour and 7-day totals.

From the persistence analysis it can be concluded that the number of days with prevailing extreme precipitation or convection-
favoring conditions during STE16 was extraordinary, but not unique. This conclusion, however, depends on the proxy considered. For the precipitation severity index $PS$ based on the 99.9% percentiles, for example, it was found that a 10-day cluster as observed during STE16 has a probability of only 0.3%. Compound events with low stability and weak mid-troposphere flow estimated from soundings are rare, but occurred several times during C20/21. A cluster with a length of 13 days for these conditions, for example, has a probability between 0.1 and 0.2% in southern Germany. The total area affected during 13 days,
where these compound events prevailed, however, was unique in 2016 and never occurred with that extent during the last half century.

Large-scale weather patterns dominating STE16 can be best described by both the objective weather type classification OWLK and two specific convection-based classification schemes. Even though the circulation pattern Low Central Europe (*Tief Mitteleuropa, TM*) according to the subjective classification of Hess and Brezowsky (HB, 1977) prevailed on 6 days,
this pattern is usually not related to severe convection (Ehmann, 2009), but to persistent advective precipitation such as during the 2002 and 2013 German floods. Hence, we decided not to investigate HB further in this context. A cluster length of 8 days exhibiting the OWLK type XX..M (where .. denote either a cyclonal or anticyclonal circulation pattern in 950 / 500 hPa, respectively) was absolutely unique and has never occurred before since 1979 (start of the OWLK calculation by DWD). The code XX here mirrors the weak wind speed in the lower troposphere as peculiarity of STE16, while M reflects the relatively
high humidity. However, OWLK has not been designed especially for the detection of conditions conducive to thunderstorms. Therefore, a new classification scheme (conOWLK) optimized with respect to convection was developed yielding specific convection-favoring weather patterns, which coincide with a very high probability of severe thunderstorm events. Additionally, a discriminant model (qdaOWLK) is applied, which provides a measure for a generally high convective predisposition in a less



strict sense. According to qdaOWLK, we estimated that less than 1% of all clusters termed to as *convective* exceeded a length of ten days. Thus, the 11-day cluster of STE16 has to be considered as highly exceptional.

A potential weakness of our research is that the examinations mainly rely on gridded REGNIE 24-hour totals and different proxies for convection, which both have a low spatial and temporal resolution. However, high resolution data sets such as radar,

satellite or lightning data are not available over a sufficiently long period of at least 30 years. In case of REGNIE data, it has to be considered that the regionalization approach and the limited number of stations lead to a spatial smoothing of the rain fields. Consequently, local rain maxima are underestimated by REGNIE. However, this underestimation is of systematic nature and affects all years, making long-term statistics reasonable also for convective rainfall. We are aware that the proxies in terms of convective parameters and weather patterns used in this study do not allow to establish a direct link to individual convective

systems. For statistical analyses, however, these data sets have a suitable prediction skill as has been shown by various studies (e.g., Haklander and van Delden, 2003; Brooks et al., 2003; Kunz, 2007; Mohr and Kunz, 2013).

For the question of whether climate change was a driver of STE16, as it was frequently asked in the aftermath by the media, it is important to note that the severe thunderstorms were triggered in an environment with moderate temperatures around 20–25°C. Thus, the potential relation between temperature increase, moisture increase and a shift in the distributions of CAPE

or SLI as shown, for example, by Mohr and Kunz (2013), does not apply for the STE16 event. In our opinion, much more matters the large annual and interannual variability of convective activity across Germany and Europe visible in the time series of all proxies investigated. The drivers of this variability, however, are not yet well understood.

In the next step we intend to scrutinize the reasons that are most decisive for the temporal variability of severe convective storms and related atmospheric conditions, and to separate among dynamical and thermodynamical processes. First results

showing a clear relation between thunderstorm days in several European regions and the North Atlantic Oscillation (NAO) Index are promising.

*Acknowledgements.* The Center for Disaster Management and Risk Reduction Technology (CEDIM) is an interdisciplinary research center in the field of disaster management funded by the Karlsruhe Institute of Technology (KIT). The authors thank the German Weather Service (DWD), Siemens (BLIDS) and the Integrated Global Radiosonde Archive (IGRA) for providing different observational data sets. The

coastDat2 simulations were performed by Helmholtz-Zentrum Geesthacht (HZG; thanks to Beate Geyer) using the National Centers for Environmental Prediction / National Center for Atmospheric Research global reanalysis (NCEP/NCAR1). We acknowledge support by German Research Foundation (DFG), the Helmholtz Climate Initiative REKLIM and Open Access Publishing Fund of KIT.





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





**Table 1.** Classification results during STE16 based on OWLK and conOWLK using the character coding scheme defined in the text (Sect. 2.3 and 2.4).

| date | OWLK | conOWLK |
|---|---|---|
| 26 May | SWCCM | XDXD |
| 27 May | SWCCM | WMID |
| 28 May | SWCCM | WMIL |
| 29 May | SECCM | WMIL |
| 30 May | XXCCM | WMID |
| 31 May | XXCCM | WMXD |
| 01 June | XXCCM | WMXL |
| 02 June | XXCCM | WMID |
| 03 June | NECCM | WMIL |
| 04 June | XXCCM | WMIX |
| 05 June | XXAAM | WMIL |
| 06 June | XXAAM | WXID |
| 07 June | XXAAM | WDID |
| 08 June | NWCAM | WMIL |
| 09 June | NWACD | CDXD |





**Table 2.** Maximum observed 24-hour totals $R_{24h}$ (in mm) at any rain gauge of DWD (name and coordinates) for each day in STE16, beginning of the record and corresponding return period $t_{RP}$ (in years). Different durations $D$ are shown (if available) together with the related heavy rainfall criterion $N_{cr}$ (in mm).

| date | station [coordinates] | data since | $R_{24h}$ | $t_{RP}$ | duration $D$ | $N_{cr}$ |
|------|----------------------|-----------|-----------|----------|--------------|----------|
| 26 May | Bitburg [49.98°N 6.53°E] | 1951 | 37.8 | $< 5$ | in 10 h (28.3 mm in 6 h) | 54.8 (42.4) |
| 27 May | Eppendorf [50.80°N 13.24°E] | 1951 | 47.4 | $< 5$ | | |
| 28 May | Siegen(Kläranlage) [50.85°N 8.00°E] | 1931 | 59.0 | 10–15 | in 4 h | 34.6 |
| 29 May | Gundelsheim [49.28°N 9.16°E] | 1888 | 122.1 | $> 200$ | | |
| 30 May | Kall-Sistig [50.50°N 6.52°E] | 1947 | 63.5 | 20–25 | in 13 h (53.2 mm in 5 h) | 62.4 (38.7) |
| 31 May | Simbach/Inn [48.27°N 13.02°E] | 1951 | 74.6 | 5–10 | 42.8 mm in 6 h | 42.4 |
| 01 June (*) | Hamminkeln-Mühlenrott [51.72°N 6.58°E] | 1931 | 120.3 | $> 200$ | | |
| 02 June | Wernigerode [51.84°N 10.77°E] | 1951 | 61.0 | 10–15 | 60.9 mm in 5 h | 38.7 |
| 03 June | Hohenpeißenberg [47.80°N 11.01°E] | 1801 | 61.3 | 5–10 | in 6 h | 42.4 |
| 04 June | Lenggries (Sylvenstein) [47.69°N 11.57°E] | 1931 | 87.5 | 5–10 | | |
| 05 June | Karsdorf [50.94°N 13.70°E] | 1978 | 65.7 | 5–10 | | |
| 06 June | Frankenblick-Mengersgereuth-Hämmern [50.39°N 11.13°E] | 1969 | 44.5 | $< 5$ | 44.1 mm in 1 h | 17.3 |
| 07 June | Durbach-Ebersweier [48.50°N 7.99°E] | 1951 | 62.3 | 10–15 | in 4 h (52.0 mm in 1 h) | 34.6 (17.3) |
| 08 June | Fellbach [48.81°N 9.27°E] | 1941 | 67.8 | 25–30 | | |
| 09 June | Neureichenau-Duschlberg [48.79°N 13.73°E] | 1931 | 38.7 | $< 5$ | | |

(*) Malfunction at Simbach/Inn on 01 June 2016, 08–11 UTC, complementation with data of the corresponding grid point of radar data yields approximately $R_{24h} \approx 120$ mm ($\approx 90$ mm in 6 h) and a return period $t_{RP}$ of 135–140 years; Heavy rain criterion $N_{cr}$ for 24-hour (6-hour) totals: 84.5 mm (42.4 mm), fulfilled in both cases.

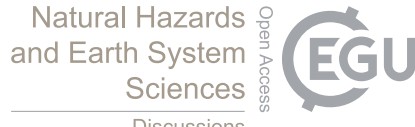



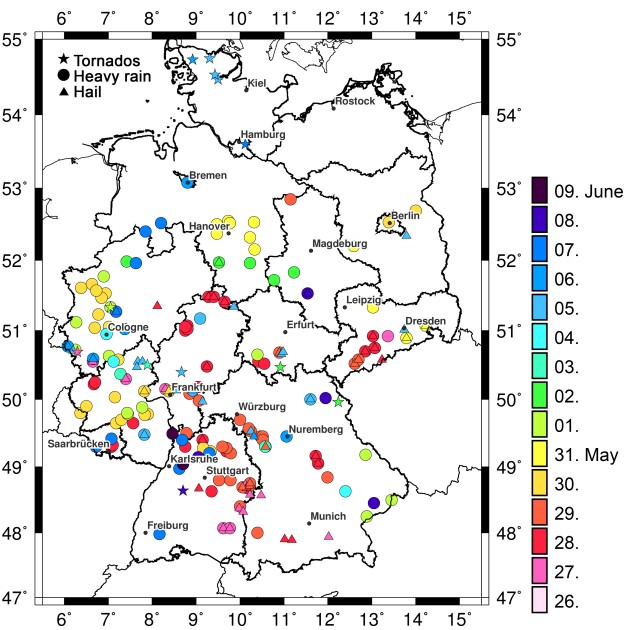

**Figure 1.** Phenomena associated with severe convective storms between 26 May and 09 June 2016 collected from various sources of information (ESWD, newspaper articles, weather services; heavy rain ●, hail △, and tornado ⋆)

.





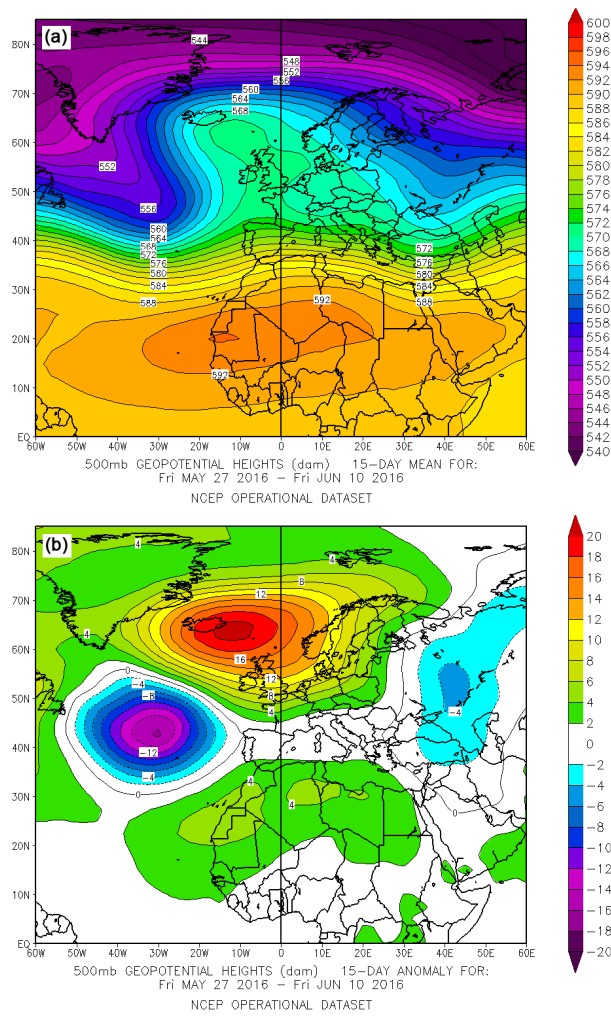

**Figure 2.** 500 hPa geopotential height over Europe, (a) averaged between 27 May and 10 June 2016 and (b) departure from normal 1979 – 2005 (Image source: NOAA/ESRL Physical Sciences Division, Boulder Colorado, http://www.esrl.noaa.gov/psd).





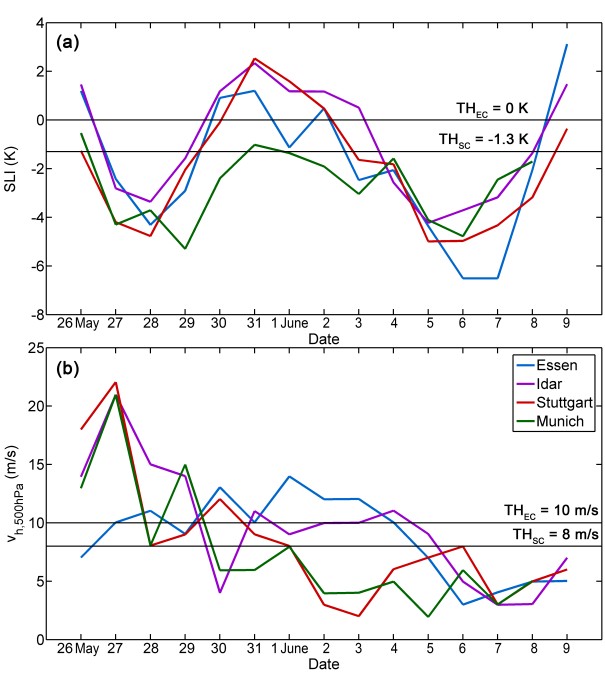

**Figure 3.** Time series of (a) Surface Lifted Index (SLI) and (b) horizontal wind speed in $500\,\mathrm{hPa}$ ($v_{\mathrm{H,500hPa}}$) at $12\,\mathrm{UTC}$ at four German sounding stations during STE16; including the thresholds (TH) defined in Section 4.3.





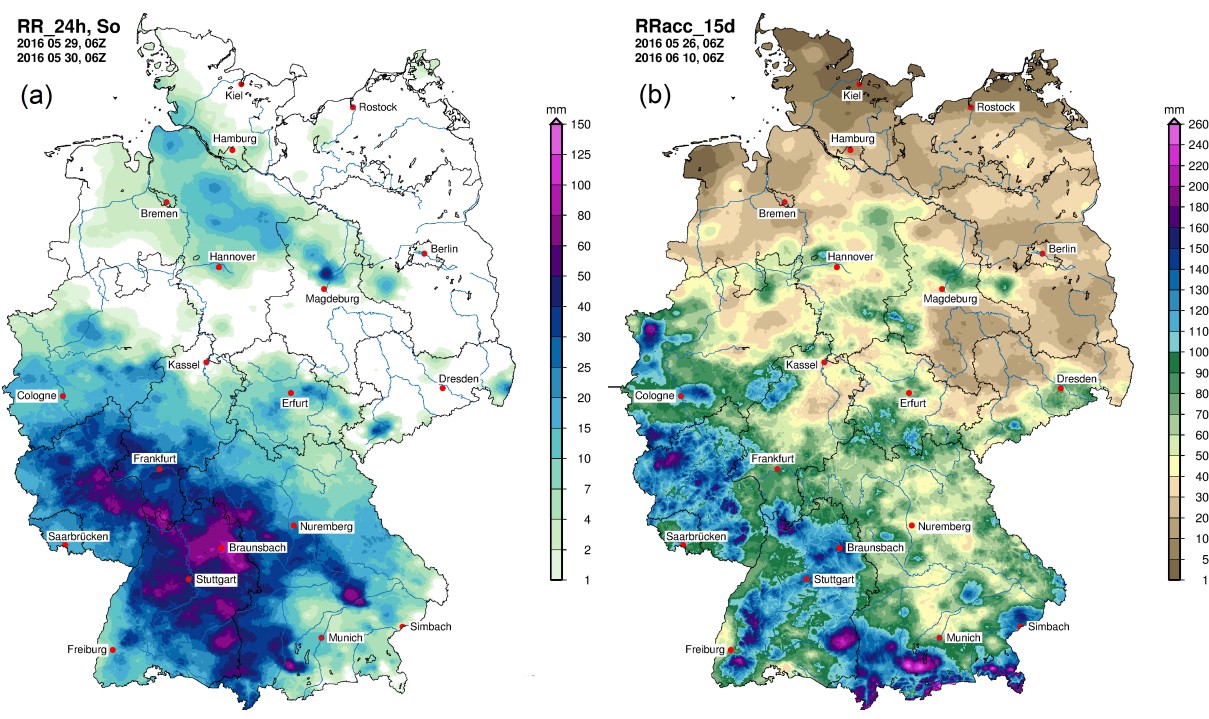

**Figure 4.** Gridded rain totals (REGNIE) over (a) 24 hours for 29 May 2016 and (b) 15 days (26 May to 09 June 2016, STE16).





**Figure 5.** Spatial distribution of maximum return periods of (a) 24-hour and (c) 7-day totals and (b, d) the day of occurrence during STE16 derived from REGNIE data; return periods are estimated with respect to the control period C20/21 (SHY).





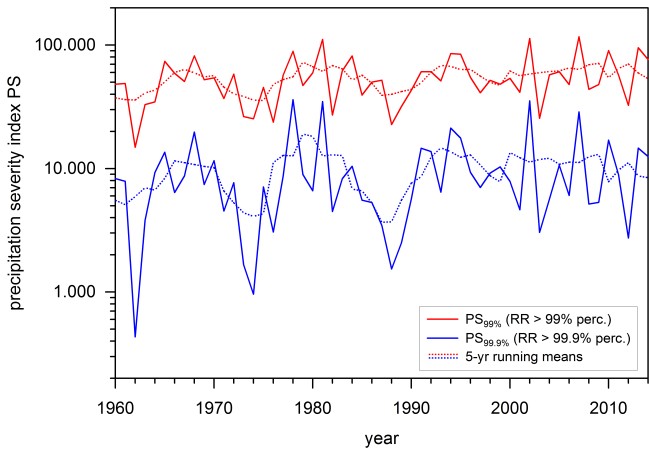

**Figure 6.** Time series of the annually accumulated precipitation severity index $PS_\xi$ for all REGNIE grid points of the investigation area including 5-year running mean during C20/21.

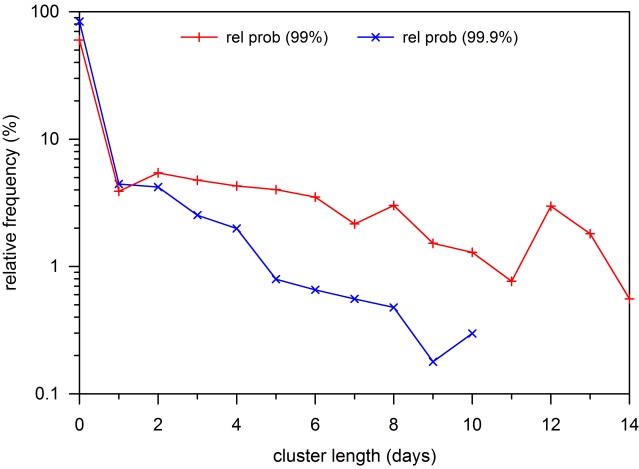

**Figure 7.** Relative frequency of clusters of consecutive days exceeding the threshold of $PS_\xi$ for the 99 and 99.9% percentiles during C20/21.




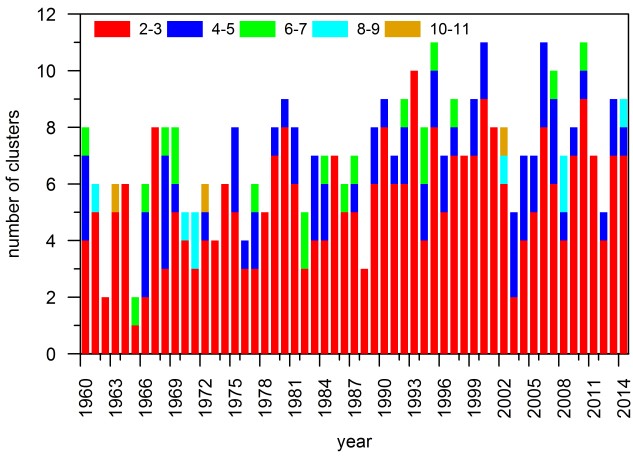

**Figure 8.** Time series of different cluster lengths with respect to consecutive days exceeding the threshold of $PS_{99.9}$ during C20/21.

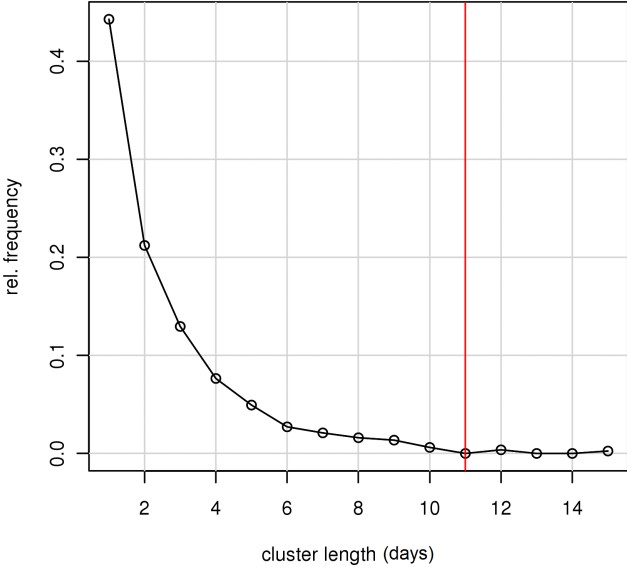

**Figure 9.** Same as Fig. 7, but for days classified as *convective* according to quadratic discriminant analysis; red: cluster length observed during STE16.





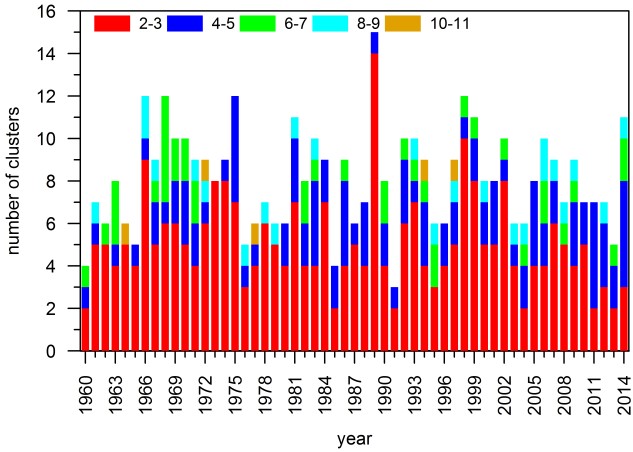

**Figure 10.** Same as Fig. 8, but with respect to consecutive days classified as *convective* according to quadratic discriminant analysis during C20/21.

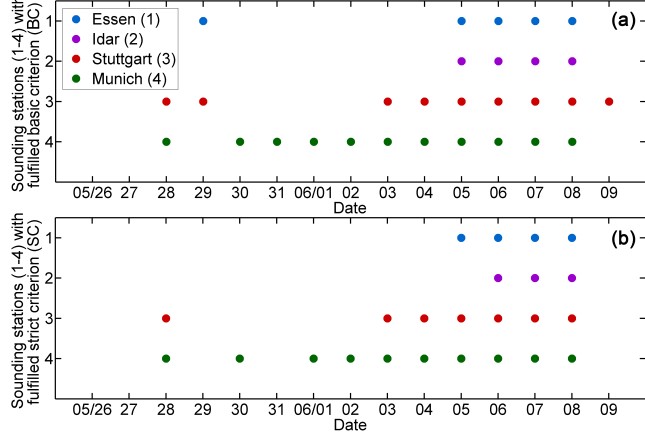

**Figure 11.** Time series of compound low stability and weak flow events at 12 UTC during STE16 at four German sounding stations for days, on which (a) the basic (BC) and (b) the strict criterion (SC) is fulfilled (represented by the dots).





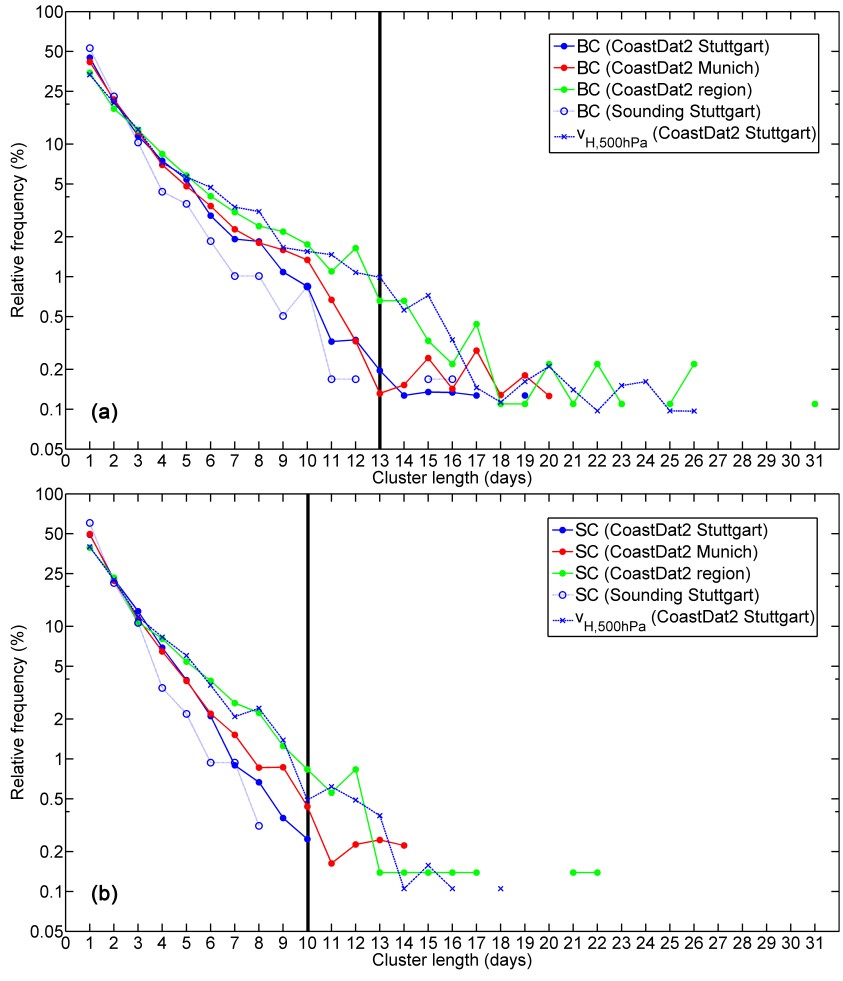

**Figure 12.** Same as Fig. 7, but for days fulfilling the (a) BC and (b) SC criteria for compound events with low stability and weak flow at the sounding station of Stuttgart (blue dotted line), $3 \times 3$ grid point averages near Stuttgart (blue solid line) and Munich (red solid line), and for days with a certain spatial extent ($A_c$; green solid line). Indicated are also clusters for wind speed near Stuttgart according to the BC/SC criteria (light dotted line). Cluster lengths during STE16 are marked as black vertical lines. See text for further details.