# Peer review of "Exceptional sequence of severe thunderstorms and related flash floods in May and June 2016 in Germany. Part I: Meteorological background"

_Natural Hazards and Earth System Sciences, 2016_

## Referee Comment (RC1) · Anonymous Referee #1 · 19 Oct 2016

Comments on the article nhess-2016-275

The paper investigates the meteorological characteristics of a severe thunderstorms episode that affected Germany and central Europe during 15 days in the spring 2016. Interesting diagnostics are used such as a rainfall severity index and weather type classification schemes. The authors point out the interaction between several ingredients : convection-favouring conditions, low stability, low wind speed. They characterise the scarcity of the episode from the point of view of the weather background It was found that this 15-days episode is very rare even if not unique. The article clearly is relevant to the field of the "NHESS" journal. The topic of the article is interesting and fully addressed. The language seems correct to me. So the paper need minor revision in the

light of the general and specific comments listed below.

General comment:

1. Does the paper address relevant scientific and/or technical questions within the scope of NHESS? Yes

2. Does the paper present new data and/or novel concepts, ideas, tools, methods or results? The paper does not really present novelties. It focuses on one particular event and makes use of several methods to characterize it.

3. Are these up to international standards? Yes, the different methods used seems useful for any meteorologist who would like to characterise the severity of this king of high impact weather event.

4. Are the scientific methods and assumptions valid and outlined clearly? Yes

5. Are the results sufficient to support the interpretations and the conclusions? Yes

6. Does the author reach substantial conclusions? Yes

7. Is the description of the data used, the methods used, the experiments and calculations made, and the results obtained sufficiently complete and accurate to allow their reproduction by fellow scientists (traceability of results)? Yes

8. Does the title clearly and unambiguously reflect the contents of the paper? Yes

9. Does the abstract provide a concise, complete and unambiguous summary of the work done and the results obtained? Yes

10. Are the title and the abstract pertinent, and easy to understand to a wide and diversified audience? Yes

11. Are mathematical formulae, symbols, abbreviations and units correctly defined and used? If the formulae, symbols or abbreviations are numerous, are there tables or appendixes listing them? Yes

12. Is the size, quality and readability of each figure adequate to the type and quantity of data presented? Yes

13. Does the author give proper credit to previous and/or related work, and does he/she indicate clearly his/her own contribution? Yes

14. Are the number and quality of the references appropriate? No comparison with studies of the same kind is provided. It would be interesting to add some references where this kind of approach is adopted.

15. Is the overall presentation well structured, clear and easy to understand by a wide and general audience? Yes

16. Is the length of the paper adequate, too long or too short? I seems adequate.

17. Is there any part of the paper (title, abstract, main text, formulae, symbols, figures and their captions, tables, list of references, appendixes) that needs to be clarified, reduced, added, combined, or eliminated? I don't think so

18. Is the technical language precise and understandable by fellow scientists? Yes

19. Is the English language of good quality, fluent, simple and easy to read and understand by a wide and diversified audience? I think so.

20. Is the amount and quality of supplementary material (if any) appropriate? No supplementary material

Specific comment : I have found only one typing error page 5 line 7. Please replace "beetween" by "between"
* * *

---

## Referee Comment (RC2) · Anonymous Referee #2 · 13 Nov 2016

The paper proposed by D. Pipper and co-authors gives a description of the exceptional meteorological event that occurred in May and June 2016 in Germany and led to several flash flood. This event is placed within a historical context using precipitation, radio sounding, and model data of the period 1960-2014 as well as lightning data of the period 2001-2014. The authors derived from these data several indicators of convective situations or favouring convection, and computed the probability of occurrence of such an event with a particular focus on the duration of the sequence of severe thunderstorm.

I think the paper addresses relevant scientific question within the scope of NHESS and that most of the review criteria are OK excepting few major and minor points I wrote

just below.

Major comments/questions:

- Section 2.4, page 6, about the quadratic discriminant analysis, I'm not an expert on that (in particular the step corresponding to equation (2)) but I wonder if the explanations are enough accurate to allow the reproduction by fellow scientist. This analysis consists of several steps using well referenced mathematical tools but the "parameters" are not detailed, in particular for the first step partitioning the groups of convective and non-convective days.

- Section 2.6, page 7, lines 12-13: the precipitation severity (PS) index is a concept I didn't find exactly in the paper of Schröter et al. (2015). Moreover, according to equation (8), PS gives no information about persistence. Finally, the unit of R and, more importantly, Gamma need to be specified (in meter, kilometre, or squared kilometre?) to better appreciate the values given later in section 4.1

- Section 5, lines 23-26, Reference to Hess and Brezowsky (1977) and the justification for not using it were not given before in the paper. It would be better to do that before the concluding section.

Minor comments/questions:

- Section 5, lines 25: The reference to Ehmann (2009) is not available online and is written in German. For this reason and the previous one, I would replace lines 23-26 by a shorter one without these references.

- Section 2.2, page 4, line 5: I think that the information about true local time may be a little confusing and not totally useful.

- Section 2.6, page 7, line 7-8: Using Wussov criterion instead of the exceedance of appropriate percentiles implies that the criterion used is specific to the German climate. I recommend a sentence to precise that.

- In the reference list, the name of the journal (Atmospheric Research) is missing for Brooks et al. (2003).

---

## Author Comment (AC1) · 22 Nov 2016

Dear Referee,

thank you very much for your work and the valuable comments how to improve the scientific quality of our manuscript. Please find below our reply to the individual points, marked with an "AC" (author's comment).

Best regards, David Piper on behalf of all co-authors

Response to the referee comments: Referee #1:

The paper investigates the meteorological characteristics of a severe thunderstorms

episode that affected Germany and central Europe during 15 days in the spring 2016. Interesting diagnostics are used such as a rainfall severity index and weather type classification schemes. The authors point out the interaction between several ingredients: convection-favouring conditions, low stability, low wind speed. They characterise the scarcity of the episode from the point of view of the weather background It was found that this 15-days episode is very rare even if not unique. The article clearly is relevant to the field of the "NHESS" journal. The topic of the article is interesting and fully addressed. The language seems correct to me. So the paper need minor revision in the light of the general and specific comments listed below

a) The paper does not really present novelties. It focuses on one particular event and makes use of several methods to characterize it.

AC: Several new methods are applied in the context of our study. For instance, the method we applied for persistence analysis, albeit based on a familiar approach (see 1b), represents a new concept, which facilitates the appropriate treatment of long series of e.g. convective days inclosing sporadic non-event days. Furthermore, we developed an objective weather type classification optimized for the detection of days prone to severe convective events. In the revised version of the manuscript, we will emphasize the novel aspects of our research.

b) No comparison with studies of the same kind is provided. It would be interesting to add some references where this kind of approach is adopted.

AC: This is a good point. The fundamental approach our method is based on, i.e. counting the number of consecutive days either classified as "yes" in a binary sense or passing a certain threshold, has been used in literature before and should definitely be cited. We will add some references.

c) I have found only one typing error page 5, line 7. Please replace "beetween" by "between".

AC: Thank you for close reading. We overlooked this error.

Dear Referee,

thank you very much for your work and the valuable comments how to improve the scientific quality of our manuscript. Please find below our reply to the individual points, marked with an "AC" (author's comment).

Best regards, David Piper on behalf of all co-authors

Response to the referee comments: Referee #2:

The paper proposed by D. Pipper and co-authors gives a description of the exceptional meteorological event that occurred in May and June 2016 in Germany and led to several flash flood. This event is placed within a historical context using precipitation, radio sounding, and model data of the period 1960-2014 as well as lightning data of the period 2001-2014. The authors derived from these data several indicators of convective situations or favouring convection, and computed the probability of occurrence of such an event with a particular focus on the duration of the sequence of severe thunderstorm. I think the paper addresses relevant scientific question within the scope of NHESS and that most of the review criteria are OK excepting few major and minor points I wrote just below.

a) Section 2.4, page 6, about the quadratic discriminant analysis, I'm not an expert on that (in particular the step corresponding to equation (2)) but I wonder if the explanations are enough accurate to allow the reproduction by fellow scientist. This analysis consists of several steps using well referenced mathematical tools but the "parameters" are not detailed, in particular for the first step partitioning the groups of convective and non-convective days.

AC: You are right; we will explain the partitioning step more in detail.

b) Section 2.6, page 7, lines 12-13: the precipitation severity (PS) index is a concept I didn't find exactly in the paper of Schröter et al. (2015). Moreover, according to

equation (8), PS gives no information about persistence. Finally, the unit of R and, more importantly, Gamma need to be specified (in meter, kilometre, or squared kilometre?) to better appreciate the values given later in section 4.1.

AC: We will overwork this paragraph according to the comments. Since the concept of persistence will be addressed in the next Section 2.7, we will delete the reference to persistence and annual variability here. For the quantification of PS it is necessary to convert all units to either $m^2$ or $km^3$. In the former version, there was a flaw in the equation; furthermore, since R is in mm ($=l/m^3$) and Gamma is in $km^2$, PS is in $m^3$ (water). This will be corrected in the manuscript and in Figure 6.

c) Section 5, lines 23-26, Reference to Hess and Brezowsky (1977) and the justification for not using it were not given before in the paper. It would be better to do that before the concluding section.

AC: We agree and decided to move this statement to the method section.

d) Section 5, lines 25: The reference to Ehmann (2009) is not available online and is written in German. For this reason and the previous one, I would replace lines 23-26 by a shorter one without these references.

AC: We will follow this suggestion.

e) Section 2.2, page 4, line 5: I think that the information about true local time may be a little confusing and not totally useful.

AC: We will delete this information

f) Section 2.6, page 7, line 7-8: Using Wussov criterion instead of the exceedance of appropriate percentiles implies that the criterion used is specific to the German climate. I recommend a sentence to precise that.

AC: We will add a statement on that. Note, however, that this criterion is only used to estimate the severity of totals observed at selected stations in Table 2.

[Figure]

g) In the reference list, the name of the journal (Atmospheric Research) is missing for Brooks et al. (2003).

AC: Thank you for close reading. We will correct that.